# A Qualitative Study: Mothers’ Experiences of Their Child’s Late-Onset Pompe Disease Diagnosis Following Newborn Screening

**DOI:** 10.3390/ijns8030043

**Published:** 2022-07-19

**Authors:** Kaylee Crossen, Lisa Berry, Melanie F. Myers, Nancy Leslie, Cecilia Goueli

**Affiliations:** 1Division of Human Genetics, Cincinnati Children’s Hospital Medical Center, 3333 Burnet Ave, Cincinnati, OH 45229, USA; lisa.berry@cchmc.org (L.B.); melanie.myers@cchmc.org (M.F.M.); nancy.leslie@cchmc.org (N.L.); cecilia.goueli@cchmc.org (C.G.); 2College of Medicine, University of Cincinnati, 3230 Eden Ave, Cincinnati, OH 45267, USA; 3Genetic Center, Akron Children’s Hospital, 215 West Bowery Street, Level 5, Akron, OH 44308, USA

**Keywords:** Pompe disease, late-onset Pompe disease, newborn screening, pre-symptomatic patients, parent perspectives, medical management

## Abstract

Pompe disease was added to the United States recommended uniform screening panel in 2015 to avoid diagnostic delay and implement prompt treatment, specifically for those with infantile-onset Pompe disease (IOPD). However, most newborns with abnormal newborn screening (NBS) for Pompe disease have late-onset Pompe disease (LOPD). An early diagnosis of LOPD raises the question of when symptoms will arise which is challenging for parents, patients, and providers managing an LOPD diagnosis. This study aimed to characterize mothers’ experiences of their child’s LOPD diagnosis and medical monitoring. A qualitative descriptive approach was chosen to gain an in-depth understanding of parental experiences. Eight mothers were interviewed about their experiences with positive NBS and diagnosis, experiences with living with the diagnosis, and experiences with medical monitoring. Interview transcripts were analyzed through conventional content analysis. Negative emotions like fear were more frequent with communication of NBS results. Participants expressed uncertainty surrounding age of symptom onset and the future. The medical monitoring experience increased worry but participants expressed that being vigilant with management reassured them. Parental emotions shifted to thankfulness and reassurance with time and education. These findings can provide guidance to providers about the psychosocial implications of receiving positive NBS results and an LOPD diagnosis.

## 1. Introduction

Pompe disease is a rare genetic disorder which affects 1 in 40,000 individuals. It occurs when two pathogenic variants in the *GAA* gene cause a deficiency of the lysosomal enzyme acid alpha-glucosidase, causing an accumulation of glycogen in muscle cells. The two disease sub-types differ by age of onset and presentation. Infantile-onset Pompe Disease (IOPD) occurs within the first year of life, and symptoms include cardiomegaly, respiratory failure, and hypotonia. Late-onset Pompe disease (LOPD) can develop any time after the age of one year with the onset of proximal muscle weakness and difficulty breathing. Late-onset Pompe disease is highly variable in age of symptom onset and disease severity, even within families [1,2].

State-based NBS is based on the Wilson and Jungner criteria (1966) which explain that screening for diseases should only occur when there is a known natural history of the disease, a recognizable early disease stage, an accurate diagnostic test, and a proven and accepted treatment plan [3]. Pompe disease was added to the recommended uniform screening panel (RUSP) in 2015 to avoid diagnostic delay and implement prompt treatment, specifically for those with infantile-onset Pompe disease (IOPD). Currently, 28 states screen for Pompe disease on NBS or have started pilot programs [4]. However, more than two-thirds of patients with Pompe disease identified from NBS are diagnosed with LOPD [5,6]. The identification of late onset forms of a disorder from newborn screen follow up testing is becoming more common. Other examples of this have been seen in screening for X-linked adrenal leukodystrophy and Fabry disease.

Diagnosing LOPD at birth cannot predict age of onset and is similar to testing minors for adult-onset conditions [1,2,7]. Early diagnoses of later-onset disorders come with benefits such as improving health outcomes by avoiding a diagnostic delay and associated uncertainty, anxiety, and turmoil for patients and families living without a diagnosis [8,9,10,11,12]. On the other hand, disadvantages discussed by health care providers and families include increased healthcare and treatment costs, insurance uncertainties, parental anxieties about the diagnosis and prognosis, medicalization of the asymptomatic child, lack of the child’s autonomy for potential adult-onset genetic testing decisions, and employability concerns [10,13,14]. However, there is limited research on the long-term benefits and drawbacks of pre-symptomatic LOPD diagnoses as NBS for Pompe disease is still fairly new.

Close monitoring of patients with Pompe disease is important to identify early symptoms and start enzyme replacement therapy (ERT) [15]. The Pompe Disease Newborn Screening Working Group recommends specific follow-up for pre-symptomatic patients diagnosed with LOPD [16]. However, as more diagnoses of LOPD are made following NBS, the impact of these recommendations on families is unknown. We interviewed participants to explore their experiences receiving a diagnosis of LOPD through NBS as well as their experiences with medical monitoring of an asymptomatic child. The aim of our study is to describe parental experiences and psychosocial implications of having a child diagnosed with LOPD due to NBS and the medical monitoring recommended for their presymptomatic child.

## 2. Materials and Methods

### 2.1. Recruitment

We recruited participants from Cincinnati Children’s Hospital Medical Center (CCHMC), outside institutions in Ohio and Kentucky, and a Pompe Facebook group. Eligibility criteria included primary caregivers of children with LOPD diagnosed following NBS who speak English. Exclusion criteria included parents of children with LOPD on ERT and parents of children who have Pompe disease related cardiac manifestations such as cardiomegaly and cardiomyopathy. LB identified patients followed at the STAR Center for Lysosomal Disease at CCHMC who met eligibility criteria. We sent invitations to participate by mail and email to six families. Emails that contained our recruitment letter to be shared with eligible participants were sent to genetics providers at five other medical institutions known by the research team in Ohio and Kentucky. KC posted an invitation to participate on a Pompe disease Facebook group.

### 2.2. Interviews

The research team created an interview guide to assess parents’ experiences with NBS results, diagnoses, and monitoring their pre-symptomatic child with LOPD (Appendix A). The interview guide also included demographic questions about both the parent and their child. The interview guide was pretested with the research team and peers not related to the study to assess question quality and flow of the interview [17]. Adjustments to questions after pretesting were made as needed. KC was also trained on interviewing and probing through pretesting the interview guide with members of the research team.

Participants provided verbal consent at the start of each interview. KC conducted interviews via Skype voice calls. Interviews were audio recorded. CG was present for five interviews and MM was present in the first two interviews to observe KC and ask participants additional follow-up questions. No participants knew the interviewer before the study. The members of the research team present on each call took notes during each interview for reference and to be analyzed alongside transcripts. Interviews lasted from 31 min to 1 h and 5 min (median: 46.5 min). Families were recontacted after interviews to thank them for their time and to request their child’s genetic testing results. Recruitment ended with no additional interested participants and no new themes emerged during the coding process.

### 2.3. Analysis

Audio recordings were transcribed verbatim through Descript, an online transcription software. K.C. reviewed transcripts for accuracy. The transcripts were entered into ATLAS.ti version 8.0 for coding. A qualitative description approach was taken [18,19]. Conventional content analysis was used to code transcripts with codes based straight from the text by reading transcripts word for word [20]. K.C. and C.G. created a codebook that included deductive codes based on questions in the interview guide and inductive codes that emerged from the transcripts. Data collection and data analysis occurred simultaneously (Appendix A) [20]. A coding plan was made that included adding the deductive and inductive codes to transcripts as first cycle coding, combining codes post first cycle coding, and creating categories and themes in second cycle coding. K.C. and C.G. coded the first transcript together as a training session. Remaining transcripts were coded independently by K.C. K.C. sent C.G. the transcripts after first cycle coding with highlighted quotes. C.G. reviewed the coded transcripts for coding agreement. K.C. and C.G. met weekly to discuss the coded transcripts and addition of new codes, as well as come to consensus on coding disagreements. Through conventional content analysis, codes were grouped into categories by K.C. and L.B. through second cycle coding [20]. Findings represent a direct descriptive summary of the categories.

## 3. Results

### 3.1. Participants

Eight participants were interviewed. Six of the eight participants were recruited from a Pompe Facebook group and two were recruited from CCHMC. All eight participants were participants aged between 29 and 38 years old, married, Caucasian/white, and had at least some college education (Table 1). Their children with LOPD diagnosed following NBS screening ranged from three months to three years at the time of interviews (Table 2).

The results of the qualitative data collected from this study were organized into the following categories: participants’ experiences with newborn screening and diagnosis, living with a diagnosis, medical management, and parental suggestions.

### 3.2. Experiences with NBS and Diagnosis

#### 3.2.1. Communication of NBS and Diagnosis

Seven participants learned of their child’s initial abnormal NBS from their child’s primary care provider (PCP)/pediatrician or a nurse (Table 3). One participant reported learning the NBS results from the health department (Table 3). All participants learned results from a phone call within a few days to two weeks after leaving the hospital after the child’s birth. These families subsequently met with geneticists and genetic counselors for additional genetic testing.

#### 3.2.2. Waiting between NBS Results Notification and Diagnosis

The wait for the initial genetic appointments, additional information, and the specific diagnosis after receiving the abnormal NBS were all described as sources of stress and anxiety. When participants reported on the way they coped, they described focusing on their newborn and other children. The waiting time for the return of confirmatory testing and diagnosis ranged from a few days after the initial call to a few months to receive genetic testing results.

“*The waiting game is so hard just to figure out if it’s late-onset or if it’s [infantile]-onset. I think that waiting game just about did me in. It was very, very hard to do….*”Participant 1

During periods of waiting, participants main source of information was from Google. Participants stated that when they received the initial news, they were specifically advised not to consult Google, but every parent reported they Googled Pompe disease which produced devastation, fear, and uncertainty.

“*I know [the nurse] told me not to Google and I’m so grateful that she told me that, but I had idle hands and an anxious heart, and I had to.*”Participant 1

“*We Googled Pompe disease and literally the first thing that comes up is these babies don’t live past their first birthday and obviously that was super heartbreaking. I don’t think I’ll ever forget that moment of seeing that and thinking that’s the possibility.*”Participant 8

#### 3.2.3. Emotions

Participants provided descriptive examples of the emotions that came with the initial news about the NBS result and during the waiting period between NBS results and initial visits and molecular test results (Table 4). All participants experienced uncertainty about which form of Pompe disease their child had, which produced anxiety. The stories from participants described trauma from the news, stress related to the possibility that their child could have a genetic disease, devastation related to the unexpected information about the newborn screen, and fear of the future. A period of sadness was described by some when they received the diagnosis of LOPD. However, all participants felt relief when their child was diagnosed with LOPD rather than IOPD.

Three participants reported they felt that being postpartum at the time of the newborn screen call and during the period of waiting for a first visit exacerbated their feelings of grief and stress.

“*It was an extremely stressful time when I first found out, especially since you’re extremely hormonal after you give birth in the first place and then to basically add to my plate that something is wrong with my child was a lot*”.Participant 5

### 3.3. Experiences with Living with the LOPD Diagnosis

#### 3.3.1. Uncertainty

Participants expressed uncertainty living with the diagnosis, specifically in terms of symptom onset and the variability of LOPD. Participants also expressed uncertainty about the future, including finances, insurance coverage for treatment, and their child’s career (Table 5).

#### 3.3.2. Emotions Related to Diagnosis

Once a diagnosis of LOPD was made, participants described many emotions relating to living with the diagnosis (Table 6). Participants expressed gratitude that their child was diagnosed early, but there were also psychosocial implications. They experienced anticipatory grief and worry related to development and symptom onset.

#### 3.3.3. Hypervigilance

Participants had a level of hyper awareness and overanalyzed their child after their child’s diagnosis. They paid extra attention to their child’s milestones and worried that a finding could be a symptom of Pompe disease.

“*… you start overanalyzing everything and that’s what I’ve been doing with her. Anytime she chokes, is that the respiratory part of the Pompe or is it just her being a normal newborn? Her legs kind of twitch every now and again, is that Pompe or is it just her being a normal newborn…. So, yeah, I guess I do look at her a little bit more fragile than I do the other kiddos.*”Participant 4

#### 3.3.4. Coping and Support

Participants utilized several support systems to cope. Many participants adjusted their focus on their family and daily life and avoided thinking about it until doctor visits approached. Two participants focused on trying to stay positive to cope. 

Participants valued having support systems for emotional support as well as for informational resources (Table 7). The most common support system discussed in this study by participants was the Facebook parent group as participants stated it provided hope and quick answers to questions. One participant described not finding support on social media. Participants listed family, friends, religion/faith, and other communities as some of their main sources of support. Providers also provided support in different ways such as educating themselves, addressing psychosocial concerns, and providing reassurance.

#### 3.3.5. Acceptance and Normalization

Some participants needed time to grieve before they were able to reach acceptance. Many families referred to their child as a normal kid and did not try to label their child because of their diagnosis as they gained acceptance and obtained more information about the diagnosis. 

“*He doesn’t have any of the symptoms and we’re not having to do any of the treatments, so to us, he is a normal kid who does normal things.*”Participant 5

One parent described the diagnosis as a risk factor rather than an actual diagnosis.

“*I don’t want people to treat my son differently because he’s got a disease. I do understand the importance of monitoring for the signs of the disease, but to me, I think my genetics counselor put it really well and she said it’s a risk factor for getting the disease.*”Participant 2

Three participants explained early diagnosis helped monitoring become normalized in their family.

“*I think what’s nice about being diagnosed so young is this’ll just be his normal, whereas you’re not taking a six year old out of the normal routine of life and having them go every three months or every six months. It’s just going to be normal for them.*”Participant 7

#### 3.3.6. Family Planning

One family (Prticipant 8) chose prenatal testing in a future pregnancy to feel prepared. One family (Participant 7) decided not to have more children because their experience with receiving a diagnosis of LOPD was traumatic. One family (Participant 5) interpreted a 25% recurrence risk as low and were reassured by autosomal recessive inheritance and chose to have more children.

### 3.4. Experiences with Medical Monitoring

#### 3.4.1. Telehealth Due to COVID-19

Our study took place during the COVID-19 pandemic, during which medical monitoring appointments were delayed or conducted by telehealth. Many felt telehealth was a challenge because they wanted the providers to provide a physical assessment. Specifically, the assessment of muscle tone was challenging via telemedicine. 

“*They want to do everything over the phone and through telemedicine and that’s hard. I really want somebody to put their hands on her and make sure that we’re doing everything possible that we can do right now because obviously, any mom would want that.*”Participant 4

Participants in the study reported their thoughts on the benefits and barriers to the recommended monitoring by their provider. Some parents identified with being educators to child’s clinicians regarding management recommendations and being advocates for their child.

#### 3.4.2. Benefits

Participants reported the greatest benefit of monitoring was that it helped comfort and ease their minds because monitoring would allow early detection of symptoms and initiation of treatment without a diagnostic delay.

“*It kind of eases some worries on our end of what if we miss something happening and we don’t know it. It is kind of comforting to have these consistent appointments.*”Participant 6

“*I don’t need to be constantly worrying about what’s going on and whether their CK levels are starting to be abnormal. I feel like knowing that there’s like a very specific thing that can tell when this is off, we’re going to need to start talking about therapy, ERT. It helps me to relax, so that to me is a huge benefit. Just the idea that we aren’t waiting for physical symptoms or physiological symptoms.*”Participant 8

#### 3.4.3. Drawbacks and Barriers

Participants explained medical monitoring of their asymptomatic children yielded many emotions, both positive and negative. The most common drawbacks were emotional drawbacks. Participants felt frustrated when disagreements occurred with providers. Participants reported worry, anxiety, and uncertainity between visits and about upcoming visits. The participants in this study did not report any barriers to monitoring in their particular cases.

“*When we have appointments, it gets a little anxious and our minds go places we shouldn’t let them but other than that…. You just get nervous that their levels are going to go up or his heart’s going to show enlargement or anything like that. There’s anxiety of having to start treatment because then it feels real. Right now, you go through day to day and it just does not feel real, but then whenever you have to go [to the hospital] and you walk into [the hospital], you have to run all these tests and it just kind of brings back memories and then it also reminds you that he actually has this. We could have to do treatment starting next week if his numbers aren’t how they should be.*”Participant 7

Time for medical visits was another drawback noted by participants. In particular, the amount of time they had to take off work, travel for appointments, and wait for visits.

#### 3.4.4. Parental Advocacy and Collaboration

Participants described being proactive about requesting more monitoring based on the Pompe Disease Newborn Screening Working Group recommendations from their providers. For example, one mother (Participant 3) changed providers for her three children with LOPD. 

“*We are really [our children’s] advocates, so us speaking up for children has been something that has not come naturally, but has been a needed asset for us.*”Participant 3

Participants also indicated that providers listened to their suggestions to alter medical monitoring. 

“*That’s actually us telling [our providers] what we feel and then we go off of recommendations from [other specialists] and say, are you comfortable with doing an echo and EKG every two years? Are you comfortable with seeing us every year and we would be seen yearly anyways. So it was more of us saying, okay, these are the recommendations we’re bringing to you. What do you think about this? It was more of us telling them and asking them what they thought.*”Participant 3

#### 3.4.5. Parental Suggestions

Participants were asked about changes they wanted in management. They explained they would like to see more provider education and preparedness for providers who give the NBS result and provide ongoing clinical care.

“*If something’s going to be put on newborn screening in that state, then have some kind of training [for providers] in it so that they can help the families.*”Participant 6

Participants made suggestions for delivering the initial NBS news. They suggested hearing the information from someone with more expertise in Pompe disese would be beneficial and that during the initial visits that providers should be mindful of the stress the participants are going through. Every parent said they were told not to Google Pompe disease, however, since parents will still Google the condition, they recommended that providers describe the types of information they may see on the internet. 

“*I so hope that she wouldn’t have told me something that I could have Googled because the first thing that popped up when we Googled it was that people with this diagnosis don’t live past a year or two. It’s just hard to even get through that. That’s not even our reality and it’s still hard for me to talk about.*”Participant 1

Participants made suggestions for other families who revceived a similar NBS result and LOPD diagnosis such as being proactive about monitoring, advocating for their child, and joining support groups such as the one found on Facebook with other parents who have children with Pompe disease.

## 4. Discussion

Newborn screening for conditions with both early-onset and late-onset forms, like Pompe disease, can result in challenges for families and providers. As Pompe disease is added to more states’ NBS panels and more conditions with common late onset forms are added as well, it is important to recognize the psychosocial burden of newborn screening for late onset disease. Mothers in our study described intense and complex emotions throughout the diagnosis period. Uncertainties were present after the NBS results, during the waiting period for a diagnosis, during living with an asymptomatic child, and relating to monitoring. Participants expressed more positive emotions after acceptance of the diagnosis and normalization of living with a pre-symptomatic child. The intense emotional responses such as anxiety, fear, and uncertainty were the strongest during waiting periods and during times of follow-up visits.

Parental fear and anxiety were most intense for parents during the waiting period between the initial NBS result call and meeting with the genetics team for both our study and Pruniski et al.’s who interviewed nine parents with children with IOPD or LOPD diagnosed following NBS. These authors also found that parents experienced increased anxiety, fear, uncertainty, and gratefulness in response to a diagnosis of LOPD [21]. During the waiting period, participants searched for more information on their own via the internet. They explained that Googling Pompe disease was traumatic. Participants reported seeing IOPD first on the internet; however, their child could have LOPD, be a carrier for Pompe disease, or have a pseudodeficiency. These other outcomes may not be clearly described on the internet and families may not have learned of these possibilities until their first genetics visit. 

Mostly, it was pediatrician offices that telephoned with the initial NBS result to families. Limited awareness about NBS and NBS outcomes for Pompe disease among primary care physicians has been reported [22]. Continually updated medical education programs on Pompe disease and other conditions on NBS will aid in better communication of this sensitive information to families, especially since they may not be gaining appropriate information from the internet. A positive experience with initial news can help build rapport and trust between the family and health care providers [23]. 

It was also reported that telemedicine for the initial visit for an abnormal newborn screen for Pompe disease was not particularly helpful to one family where this was recommended as they were concerned that the physical exam and muscle tone would be difficult to assess. It is also possible that the barriers that go along with telemedicine such as technical difficulties with technology, building rapport, and educating families virtually may be challenging. However, telemedicine may be a good avenue to increase access to care for families that live far from a specialty provider and can decrease the wait time to be seen. Increased effort could be placed in this area, to improve the telehealth experience for these families with the caveat that some abnormal newborn screen indications are better for telemedicine while others may not be. 

### 4.1. Experiences with Living with the Diagnosis

We observed that participants were hypervigilant about their child’s development as a result of the uncertainty about symptom onset. The participants we interviewed were also uncertain about what the future would hold for their child. The increased hypervigilance may lead to medicalization of the asymptomatic child with LOPD as well as the patient living with a burden of knowledge supporting the notion of “patients in waiting” [21,24,25].

Timmermans and Buchbinder introduced “patients in waiting” when describing how NBS would increase the number of diagnosed asymptomatic patients that do not need treatment [26]. “Patients in waiting” live not knowing when or if their symptoms will arise which causes a high amount of uncertainty for pre-symptomatic patients and their families [26].

The participants in our study coped with negative emotions and uncertainty by reaching out to other families with children who have Pompe disease on social media groups. This helped participants gain a more realistic idea of what the diagnosis would look like and provided hope. Participants also reported that joining and interacting with social media groups helped make them feel like a member of the Pompe disease community. These findings contrast to other parents interviewed by Pruniski et al. who reported isolation and loneliness in social media support groups, particularly if their child had no symptoms compared to others in the group [21]. 

### 4.2. Experiences with Medical Monitoring

There is currently no consensus on medical monitoring for LOPD. The Pompe Disease Newborn Screening Working Group has published recommendations for early management of infants with IOPD and symptomatic LOPD after a positive NBS [16]. The application of guidelines can be a challenge for clinicians due to gaps in the understanding of symptomatic vs asymptomatic LOPD. There is also no consensus on determining when a patient with LOPD via NBS requires intervention such as with enzyme replacement therapy. Medical monitoring for asymptomatic children also impacted parental anxiety and uncertainties, particularly as medical visits approached. 

Participants expressed gratefulness for monitoring because it provided reassurance that their child would be able to start ERT as soon as symptoms arise. Routine follow-up and checkups have been found to provide security to “patients in waiting” [25]. Although the multiple visits and laboratory tests may help reassure participants that they are doing everything they can, they may also create unnecessary stress and anxiety, or “anticipatory grief”, as described by one participant. Recognition by providers of the psychosocial implications of medical monitoring of asymptomatic patients and collaboration between parents and providers will be important to find a balance that eases stress and anxiety for parents. 

Most of the participants in our study wanted to be as proactive as possible and found it frustrating when monitoring plans were not clear. Similarly, newborn screening for Krabbe disease (KD) has also led uncertainties for families whose children are at risk of developing later-onset KD. When New York initiated screening in 2006, monitoring plans for asymptomatic children diagnosed with KD through NBS were still evolving, which increased uncertainty for providers and families [26]. Providers suggested intensive follow-up for KD. Some families were not willing to subject their children to invasive assessments when they were asymptomatic, thus increasing risk for loss-to-follow up [5,27,28]. Following asymptomatic individuals for LOPD and other late onset disorders diagnosed early should be balanced carefully. 

## 5. Conclusions

This study informs providers of the parental experience of receiving a diagnosis of a late onset disorder on NBS. Mothers expressed balancing the emotional dynamics of receiving this diagnosis to living with diagnosis. This should be evaluated by providers so they can guide parents based on their coping style and provide the resources they need. Additionally, increased provider education and attention to recognizing and addressing the fear, anxiety and uncertainty provoked by having a child diagnosed with LOPD pre-symptomatically could benefit families impacted by the diagnosis and improve patient and provider relationships. Further research is important to understand the familial experience, especially as more disorders with late onset forms are added to NBS. Future research into what families experience, how these children grow up, and how to predict disease outcomes will benefit how providers make recommendations for following pre-symptomatic individuals and personalizing their care. Further exploration of the benefits and limitations of monitoring asymptomatic patients from the parental perspective could help inform future practice and policies regarding communicating NBS Pompe disease results and follow-up medical monitoring plans. This is important not only for LOPD but for other asymptomatic individuals diagnosed with late onset disorders as other diseases appear on the recommended uniform screening panel in the United States.

### Limitations

Our study consisted of eight participants, a small sample size, which limits generalizability to all parents with children with LOPD diagnosed because of NBS. The participants in our study were all mothers, were all white/Caucasian, and had an education level of some college or above. These findings may not be generalizable to all parents of children with LOPD diagnosed following NBS. Methods of recruitment for future studies should address the significant gap of diversity in this study. Future studies should include people of color, fathers, and those of differing economic backgrounds to understand the impact of identifying later-onset disorders on NBS.

In addition, six of the eight participants were recruited through a Pompe disease Facebook group, suggesting participants may have had a bias towards using social media as support. Specifically, the mothers in this category may be more likely to use social media for support, information, and anticipatory guidance about LOPD than other parents of children with LOPD.

Newborn screening for Pompe disease is relatively new so the asymptomatic patients were all aged three and under. Longitudinal studies that follow the natural history of children diagnosed with LOPD on NBS and the monitoring they receive over time will inform updates to management of patients with LOPD.

## Figures and Tables

**Table 1 IJNS-08-00043-t001:** Parent Demographics.

Parent Demographics		*n*
Age	Range Median	29–38 35
Gender	Female	8
Race	White/Caucasian	8
Marital Status	Married	8
Highest Education	Some College	1
	College Degree	2
	Some Graduate	1
	Graduate	4

**Table 2 IJNS-08-00043-t002:** Child Demographics.

Participant	Child’s Age	Child’s Gender	Child’s Birth Order	State Screened	Child’s Genotype
1	5 months	Male	2 of 2	KY	c.-32-13T>G homozygous
2	3 months	Male	2 of 2	MO	c.-32-13T>G homozygous ^c^
3	2 years	Female	6 of 7	WI	c.-32-13T>G and 2242 dupG
4	2 months	Female	3 of 3 ^a^	KY	c.-32-13T>G and c.2481+110_2646+39del
5	2 years	Male	1 of 1	KY	c.-32-13T>G homozygous ^b^
6	1 year	Male	2 of 2 ^a^	NJ	c.-32-13T>G homozygous ^c^
7	9 months	Male	3 of 3	Il	Unknown
8	3 years	Female	1 of 2	Il	c.-32-13T>G homozygous

^a^ includes half sibling; ^b^ taken from parent’s results; ^c^ by parental report.

**Table 3 IJNS-08-00043-t003:** Parent reports of learning about NBS result.

Participant	Caller with NBS Result	NBS Result Timing after Birth
1	Pediatric Nurse	8 days
2	Pediatric Nurse	3–4 days
3	Pediatrician	7 days
4	Primary Care Physician and member of genetics team	7 days
5	Pediatrician	2 weeks
6	Pediatrician	7 days
7	Primary Care Office	10 days
8	Health Department	7 days

**Table 4 IJNS-08-00043-t004:** Emotions related to the initial NBS result, waiting period, and diagnosis of LOPD.

Emotion	Quotes
Grief	*“**At first, I feel like I went through all the stages of grief, probably three times over.”* Participant 1.
Fear	*“We were terrified because they just said Pompe disease…They didn’t say what types and the more we Googled, the more we found out that the life expectancy isn’t long. We’re looking at this baby that they just said is completely fine, healthy, and everything was great and then had this just hanging over us. It was probably the hardest week of our lives, to be honest.”* Participant 6. *“It was a rough go for a bit because you have the happiness of having a child and that happiness just drops into fear.”* Participant 6.
Traumatized	*“We didn’t really have a super great experience at first because it’s like our doctor’s office was just kind of like ‘I’m sorry we don’t know anything about it, but your child could die in a year’… I know they do not have to know everything, but they did not know anything before they called us. So, it was kind of traumatic, to say the least.”* Participant 7.
Devastation	*“We were just constantly devastated in the first month”.* Participant 8.
Relief	*“I guess obviously very relieved that if you had to have some type of the disease, that was the best kind to have, but also just heartbroken he’d have to go through this in his life.”* Participant 6. *“I remember feeling relieved that it was late-onset Pompe, but at the same time, thinking now I’m going to have to worry about this for the rest of her life.”* Participant 8.

**Table 5 IJNS-08-00043-t005:** Uncertainties related to participants having a child living with an LOPD diagnosis.

Uncertainty Theme	Quotes
Symptom Onset	*“I think it is kind of hard since it’s such a spectrum and it’s such a wide spectrum. It’s kind of hard because I wanted to plant my feet firmly on what it was going to look like.”* Participant 1.
Future Insurance Coverage of Treatment	*“I think maybe the biggest worry I have is when we get there is insurance going to [cover treatment]. What’s their criteria for making a payment on a very, very expensive drug?”* Participant 2.
College and Career Choices	*“You want to say to your kid you can be whatever you want, but then in the back of your head you have that nagging of what if they make a career choice they can’t [physically] do, then what? You don’t want them to struggle.”* Participant 6.

**Table 6 IJNS-08-00043-t006:** Emotions related to participants having a child living with an LOPD diagnosis.

Emotion	Quotes
Grief	*“I just feel like, someone used the word, anticipatory grief, when we’re going to see symptoms and how we’re going to handle it when it gets here kind of a thing.”* Participant 1.
Sadness	*“I’m not devastated anymore. I think me and my husband both say that we have days where it’s still kind of hard, like it hits us and we’re just having a bad day about it.”* Participant 1.
Gratitude	*“We’re thankful that we have the best end of the deal versus the other side, which I couldn’t imagine having the infantile-onset, but still we would still fight and do whatever it takes to get the best care that we can.”* Participant 4.

**Table 7 IJNS-08-00043-t007:** Support systems reported by participants through diagnosis period and after.

Support System	Quotes
Facebook Parent Group	*“I got in contact with all those moms on the Facebook page, that helped probably more than anything, just knowing that kids can be normal and okay with the disease. Then also seeing how variable it is, well it’s just so all over the board what it can do to your body, so I don’t know. It’s a scary thing, but I feel better with the knowledge I have now.”* Participant 2. *“Its really nice to jump onto the group and just quickly ask a question and you get 54 responses or comments to this one question…it makes you feel like it’s a bigger community than what we are because we are such a rare thing here.”* Participant 3. *“I went on Facebook and got on some different support groups. That really seemed to help the most because getting people’s actual life experiences was much better than Google.”* Participant 4. *“Whenever you get on the Facebook groups, you’re like, oh my gosh, there’s pictures of these little boys or girls doing so good and that just gives you hope.”* Participant 7.
Family/Friends	*“I feel like it’s really important to have your family close to you in things like this. Just because you have your older kids who need picked up from school. Like just getting everything lined up whenever you do have appointments and things like that. And then emotionally, when you go through it all, it’s nice to have everybody there and just some people to lean on when you go through it.”* Participant 7.
Religion/ Spirituality	*“I had posted on my personal Facebook page asking my friends and family for prayers for her. We were going to something and we were waiting; that we received a diagnosis and we were waiting on more details about the diagnosis. I said that we wanted everybody to pray for us.”* Participant 1.
Providers	*“Just knowing that [my pediatrician] had done his homework on this to learn more, made me feel very supported.”* Participant 1. *“… I was by myself doing all the appointments with [a social worker], so having somebody so upbeat and positive really, really, really helped me.”* Participant 3. *“Whenever we are anxious, and we get [to the research center], and we see the doctors, the doctor’s kind of ease our mind. They remind us treatment is an option. There’s some kids out there that have genetic diseases that we can’t do anything about and there’s therapy and he’s going to be okay. They reassure us that everything’s going to be fine and I guess that’s how we cope.”* Participant 7. *“I remember as clear as day the genetic counselor said. ‘Well, your job is to love her and you’re going to leave the worrying about that to the experts because we’re going to follow her and keep track of her CK levels and keep track of her development. And we’re going to be the ones to worry about that. And your job is to just love her’ I’ll never forget that either. It totally changed my perspective on the whole thing.”* Participant 8.

## Data Availability

Not applicable.

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
