# Peer review of "A Qualitative Study: Mothers’ Experiences of Their Child’s Late-Onset Pompe Disease Diagnosis Following Newborn Screening"

_2409-515X, 2022, doi:10.3390/ijns8030043_

Round 1

Reviewer 1 Report

The manuscript by Crossen et al, presents work examining parental insights and experiences in receiving a late-onset Pompe disease diagnosis through newborn screening. Overall, the manuscript is well-written and provides several observed themes that can contribute to better understanding and implementation of Pompe disease NBS. 

A couple of additions will enhance this paper:

1) Look for updated literature regarding the relative % of IOPD versus LOPD detected through NBS. The cited article is nearly 5 years old and was published shortly after Pompe disease was added to the RUSP in the US.

2) Line 44-45: Be more clear that this is referring to benefits of early diagnoses of later-onset diseases.

3) Line 60-63: Unclear sentence. Try: The aim of our study is to describe parental experiences and psychosocial implications of having a child diagnosed with LOPS due to NBS and the medical monitoring recommended for their presymptomatic child.

4) Additional analyses that would enhance the paper include, looking at time since initial diagnosis - how did parental responses compare between those that were months out from diagnosis versus those that were years. How did insurance status alter perspectives given the ongoing monitoring needs? Was there any correlation with monitoring plan and hypervigilance or anxiety to time to initial GC visit?

5) The study is done on a highly biased and small cohort (N = 8, all white, all with at least some college, all married). This needs to be discussed as a limitation of the study as well as how answers may differ from parents in lesser resourced situations. It is important for public health programs to understand impacts from all families potentially impacted, and not addressing this is a significant gap in the paper. Please add a section that addresses these limitations, potentially biased conclusions, and propose ways the community can enhance future research in this area with more diverse participants.

Reviewer 2 Report

This is an interesting study on perspectives and experiences parents receiving late onset Pompe diagnoses for their newborns.  Given the expansion of NBS programs nationally and an increased emphasis on the impact that later onset diagnoses may have on families, this manuscript is is both timely and relevant. However, while this manuscript may be appropriate for publication in IJNS, there are a number of points that should be addressed by the authors before considering publication:

Background:

1.     While there is a nice description of the recent addition of Pompe to NBS panels, the background is somewhat incomplete. A bit more detail on Pompe disease, the process by which the condition was added to the Recommended Uniform Screening Panel, and the ongoing implication of the condition at the state level would be helpful to contextualize the recent NBS changes regarding Pompe. For example, the background currently makes it sound like every state is screening for Pompe, and I am not sure that's the case yet. As of early 2021, only 23 states were currently screening for Pompe. If that has changes, please provide a reference.

2.     It would also be helpful to add some language discussing how Pompe is similar or different from other conditions on the NBS panel, which would make the need for a study like this vital. Is just the late onset variant that is key here, or are there other aspects of Pompe that are important to discuss?

Methods:

1.     Given state by state differences, it would be helpful to describe the other institutions that you recruited from in more detail if possible.

2.     What efforts were made to recruit a more diverse sample? This is especially important as we consider the potential disparities in access to ongoing follow up and support.

Results

1.     At times the results section feels more like a listing of quotes without more context. It might be beneficial to add a bit more language to better contextualize and compare the results.

2.     Did you see any significant differences between the attitudes and experiences between parents recruited online vs. through the hospital (even though this was a small number)?

Discussion/Conclusions

1.     A limitations section and language on future research directions is needed to more fully address how these data could be interpreted and the potential implications of these findings for NBS programs.

2.     Overall the discussion and conclusions section should be much clearer about what the main “take home” messages of these data should be and what kinds of policy and practice implications the finings might lead to. While these data are exploratory, a cleaner discussion and conclusion section may help to focus how these data can help address potential issues for the success of NBS for Pompe disease.

Round 2

Reviewer 2 Report

The authors have done a nice job addressing the issues identified in the first review.